# Prevalence and Preferred Niche of Small Eukaryotes with Mixotrophic Potentials in the Global Ocean

**DOI:** 10.3390/microorganisms12040750

**Published:** 2024-04-08

**Authors:** Kaiyi Dong, Ying Wang, Wenjing Zhang, Qian Li

**Affiliations:** 1School of Oceanography, Shanghai Jiao Tong University, 1954 Huashan Road, Shanghai 200030, China; 2Key Laboratory of Polar Ecosystem and Climate Change, Shanghai Jiao Tong University, Ministry of Education, 1954 Huashan Road, Shanghai 200030, China

**Keywords:** mixotroph, heterotroph, autotroph, eukaryotes, niche, competition, *Tara Oceans*, 18S rRNA gene

## Abstract

Unicellular eukaryotes that are capable of phago-mixotrophy in the ocean compete for inorganic nutrients and light with autotrophs, and for bacterial prey with heterotrophs. In this study, we ask what the overall prevalence of eukaryotic mixotrophs in the vast open ocean is, and how the availability of inorganic nutrients, light, and prey affects their relative success. We utilized the *Tara* Oceans eukaryotic 18S rRNA gene and environmental context variables dataset to conduct a large-scale field analysis. We also performed isolate-based culture experiments to verify growth and nutritional resource relationships for representative mixotrophic taxa. The field analysis suggested that the overall prevalence of mixotrophs were negatively correlated with nutrient concentrations and positively associated with light availability. Concentrations of heterotrophic bacteria as a single variable also presented a positive correlation with mixotrophic prevalence, but to a lesser extent. On the other hand, the culture experiments demonstrated a taxa-specific relationship between mixotrophic growth and nutrition resources, i.e., the growth of one group was significantly dependent on light availability, while the other group was less affected by light when they received sufficient prey. Both groups were capable of growing efficiently with low inorganic nutrients when receiving sufficient prey and light. Therefore, our field analysis and culture experiments both suggest that phago-mixotrophy for ocean eukaryotes is seemingly an efficient strategy to compensate for nutrient deficiency but unnecessary to compensate for light scarcity. This study collectively revealed a close relationship between abiotic and biotic nutritional resources and the prevalence of trophic strategies, shedding light on the importance of light and nutrients for determining the competitive success of mixotrophs versus autotrophic and heterotrophic eukaryotes in the ocean.

## 1. Introduction

Pico- and nano-sized small eukaryotes (<3–20 µm) represent the dominant microbial communities in ocean food webs. Through metabolic activities such as photosynthesis and phagocytosis, they regulate the flux of elements and energy between primary producers and consumers. Traditionally, all protists with inherent chloroplasts were considered strict autotrophs (which only photosynthesize), while those without inherited chloroplasts were regarded as strict heterotrophs (which only phagocytose). It has become increasingly evident that many of those autotrophs/heterotrophs are in fact mixotrophs, phagotrophic mixotrophs particularly, as they are capable of both photosynthesis and phagocytosis [1,2,3]. Depending on whether the organism possesses its own chloroplasts, mixotrophs can be further categorized into constitutive mixotrophs and non-constitutive mixotrophs. Constitutive mixotrophs, the focus of the current study, are normally small-sized flagellated eukaryotes that primarily feed on bacterial prey [4,5], whereas non-constitutive mixotrophs are larger-sized microplankton (20–200 µm), which graze on both eukaryotic and bacterial prey [6,7]. Due to their distinct metabolic preferences and trophic activities, those two mixotrophic groups should be studied separately. 

The relative abundance and prevalence of mixotrophs (compared to other eukaryotes) in a given system can be partially interpreted as the competition results among mixotrophs, autotrophs, and heterotrophs. Specifically, mixotrophs and autotrophs compete for inorganic nutrients (hereinafter referred to as nutrients) and light for autotrophic growth, and for bacterial prey (including heterotrophic and autotrophic bacteria) with bacterivorous heterotrophs for heterotrophic growth. Although results may vary, previous studies have suggested that the physiological performance and competitive advantage of mixotrophs can be related to these three nutritional resources of nutrients, light, and prey [8,9,10,11]. Mixotrophs are most likely to benefit from scenarios where nutrition is not sufficient to support a single trophic strategy, i.e., strict autotrophy or heterotrophy. For example, Ward (2019) proposed a conceptual model arguing that mixotrophy can effectively alleviate dissolved inorganic nitrogen deficiency in oligotrophic environments via phagocytosis of high-nitrogen bacterial prey [12]. Meanwhile, it is necessary to acknowledge that the photosynthesis and/or phagocytosis performance alone is very likely penalized for mixotrophs [13], and these tradeoffs can differentiate significantly among different species. Li et al. (2022) demonstrated distinct grazing rates and biovolume conversion rates among different mixotrophic strains [5], suggesting varying capabilities of phagotrophy and gross growth efficiencies.

In regional areas of temperate sea [14,15,16] and oligotrophic ocean [11,17,18] or lakes [19,20,21], field observations have also shown that mixotrophy can be advantageous for phytoplankton community growth, when nutrition is not ideal for autotrophy, which is mostly due to limited nutrients and/or irradiance. In other words, conditions that are unfavorable for strict autotrophy or heterotrophy could possibly select for mixotrophy on an entire community level. However, for larger spatial scales that encompass multiple ocean regions, there have been few studies looking into the broad-scale relationship between mixotrophic prevalence and environmental conditions related to nutrition availability. While acknowledging these relationships can be complex due to multi-factorial interactions among biotic and abiotic components in the food web, it is reasonable to speculate nutrients, light, and prey can play an important role determining the relative success of mixotrophy. With climate change, evidence has shown that mixotrophs may play a more important role due to their metabolic plasticity, i.e., they find it easier to adapt to changing environments [22,23,24,25]. Motivated by these findings, we are particularly interested in delineating the overall prevalence and realized niche favoring mixotrophs in comparison to strict autotrophs and heterotrophs in vast open oceans across the globe.

The broad goal of this study was to gain insights into the interrelationship between environmental conditions and trophic strategies on both a community and a species level, as well as the underlying mechanism behind it. We aimed to answer the following scientific questions: (1) What is the overall prevalence of mixotrophic eukaryotes in open oceans? (2) Are there specific environments under which mixotrophs are favored over autotrophs and heterotrophs? (3) If so, what roles do nutrients, light, and bacterial prey play in determining the relative success and fitness of mixotrophy? We used the eukaryotic 18S rRNA gene dataset from the *Tara* Oceans Expedition to assign trophic modes to each lineage/species and to assess the relationship between trophic mode composition and contextual variables. Subsequently, we ran culture experiments to confirm relationships between representative mixotrophic isolates and key nutritional resource variables. The *Tara* Oceans analysis revealed statistically important variables that are affiliated with the trophic mode prevalence, and the experimental results provided detailed and physiological evidence for it.

## 2. Materials and Methods

### 2.1. Study Regions

The data generated by the *Tara* Oceans expedition between 2009 and 2013 are particularly suited to studying the genetic and functional diversity of plankton. In this study, a total of 104 stations and 165 samples were investigated, among which all 104 sites had surface samples and 61 sites had both surface and chlorophyll maximum layer (CML) samples. A surface dataset (SUR) consisting of 104 samples was used, while the surface and chlorophyll maximum depth dataset (SUR&CML), comprising 122 (the 61 stations had both SUR and CML samples) and 165 samples (all 104 SUR samples and 61 CML samples), was generated for a different statistical analysis. These samples were collected during four different seasons (spring, summer, autumn, and winter) between 2009 and 2013 from eight oceanic regions (Northern Atlantic Ocean, Mediterranean Sea, Red Sea, Indian Ocean, Southern Atlantic Ocean, Southern Ocean, Southern Pacific Ocean, and Northern Pacific Ocean) with broad latitudinal (64.4° S–43.7° N) and longitudinal ranges (159.0° W–73.9° E). Bathymetry depths varied from 13 to 5964 km and distance to coast was between 0.9 and 1404 km.

### 2.2. Contextual Environmental Variables

We built an environmental context dataset for further statistical analysis, following these three steps. First, representative environmental variables were obtained from both the *Tara* Oceans PANGAEA data repository (http://www.pangaea.de) and a variety of publications, including Pesant et al. (2015), Picheral et al. (2017), Ardyna et al. (2017), Ibarbalz et al. (2019), Faure et al. (2019), Karlusich et al. (2022), etc. (Appendix A) [26,27,28,29,30,31]. The major approaches for retrieving these variables included in situ sensors, laboratory analysis of water samples, climatological instruments, and model simulations, following standard protocols and analytical methods (details are described in the aforementioned literature). Second, these 100 variables were manually filtered down to 77 by eliminating similar variables and less relevant ones, to reduce collinearity and the inflation of dimensionality. Examples include the fact that salinity was chosen over conductivity, and factors such as moon phase durations and current residual time, etc., were excluded. Lastly, when variables had values from multiple measurements and/or calibration/calculation methods, only the most accurate one or suitable one (for the purpose of this study) was selected. For example, median values for photosynthetically available radiation (PAR), averaged on an 8-day interval and calculated based on the attenuation coefficient and surface irradiance observed by Moderate Resolution Imaging Spectroradiometer (MODIS), were selected over a 1-day interval to avoid fluctuations caused by short-term weather conditions. Total chlorophyll *a* concentrations, derived from sensor measurements and calibrated with non-photosynthetic quenching and discrete water sample measurements, were used, among other available measurements. Additional comparisons were also conducted to confirm there were no overall significant differences between the selected measurements and other measurements for the same variable, such as 8-day averaged PAR vs. 30-day averaged PAR, total Chl *a* derived from sensor vs. water samples vs. MODIS, and nutrient concentrations derived from water samples and model simulation, etc. 

The final contextual environmental dataset comprised 29 variables (Appendix A), representing physical conditions such as temperature, salinity, density, attenuation coefficient of PAR (Kd PAR), chlorophyll maximum layer (CML), mixed layer depth (MLD), nitracline depth and euphotic depth; climatology, such as PAR and sunshine duration; biochemical conditions, such as total Chl *a*, oxygen, nutrients (NO_3_^−^&NO_2_^−^, NO_2_^−^, PO_4_^3−^, SiO_4_^−^), pH, total alkalinity, total carbon, fluorescence, particulate organic carbon (POC), particulate inorganic carbon (PIC), and bacterial abundances, including autotrophic (*Prochlorococcus* and *Synechococcus*) and heterotrophic bacteria. Note that longitudes, latitudes, distance to coasts, sampling depths, and bathymetry depths were not included in the analysis, in order to better focus on the aforementioned climatological, biochemical, and physical variables.

### 2.3. 18S rRNA Gene Abundances and Corrected Cell Abundances

Detailed genomic sample processing methods were described in de Vargas et al. (2015) [32]. Briefly, community genomic DNA was extracted from cells filtered through membrane filters with different pore sizes (0.8 µm, 5 µm, and 20 µm) after pre-filtration with nylon sieves (to retrieve different plankton communities). In this study, we focused on the 0.8–5 µm size range, which constituted the majority of eukaryotic communities in the sampled regions [32]. The eukaryotic V9-18S rRNA gene region was sequenced through Illumina HiSeq, assembled, filtered, and classified with SWARM for operational taxonomic units (OTUs). The sequence files were downloaded from the European Nucleotide Archive under project number PRJEB6610. Subsequently, we re-assigned OTUs to the lowest taxonomic rank possible, referring to the Protist Ribosomal Reference (PR^2^) database [33]. Ciliophora, Radiolaria, and Foraminifera, which were likely derived from large-sized populations (contributing a small portion of the total species), were excluded from further analysis to focus on small-sized constitutive mixotrophs. Unknown Eukaryota and a small portion of parasites (e.g., Fungi, Amoebozoa, Syndiniales, and Dinoflagellata) were excluded from further trophic annotation, and, thus, further analysis. This final gene dataset generated 37,181,755 sequences (gene abundances) and 725 lineages/species for surface samples, and 17,874,753 sequences (gene abundances) and 802 lineages/species for the CML samples, respectively (Appendix A). 

Eukaryotic communities are known to exhibit substantially different 18S rRNA gene copy numbers, varying from as low as one copy, in species belonging to Ochrophyta, to thousands of copies per cell in dinoflagellates [5,34]. Overall, cellular biovolumes have been shown to be positively correlated with 18S rRNA gene copy numbers, but certain taxa, such as dinoflagellates, tended to have particularly high copy numbers [35,36]. To correct for this bias and obtain gene-converted cell abundances, we first collected empirical data from plankton, focusing on small cells with equivalent diameters of 5–10 µm and biovolumes of 65–523 µm^3^ (Appendix A). Then, we fit a linear correlation equation between empirical 18S rRNA gene copy numbers and cellular biovolume for individual group of dinoflagellates, diatoms, and other eukaryotes (including chlorophytes, haptophytes, dictyochophytes, etc.), respectively. Finally, a correction factor (C.F.) for each group was calculated using upper limit biovolume of 65 µm^3^; that gave 59.3, 10.5, and 4.7 for dinoflagellates, diatoms, and other eukaryotes, respectively. For comparison, Martin et al. (2022) used 27.1, 4.4, and 0.9 C.F. for dinoflagellates, diatoms, and all other eukaryotes [36]. Gene abundances were converted into cell abundances by dividing gene numbers with the C.F. for each group. The corrected cell abundance was the focus for further analysis, but 18S rRNA gene abundances were also assessed occasionally for comparison purposes.

### 2.4. Assignment of Trophic Modes

In this study, we focused on the trophic strategy potentials among different eukaryotic lineages, rather than the in situ active trophic activities (which are partially associated with instantaneous conditions). Therefore, we assigned each lineage/species (when possible) into three trophic groups by searching published database and references [5,37,38], including (potential) mixotrophs (pigmented eukaryotes with phagocytosis capability), heterotrophs (eukaryotes with no inherent chloroplasts), and autotrophs (pigmented eukaryotes with no available evidence for phagocytosis). When trophic annotation was impossible at lineage or species level, their affiliated genus or family was referred to for annotation. The filtration generated a final composition of 184–186 mixotrophic, 283–313 heterotrophic, and 303–377 autotrophic lineages/species for the SUR and SUR&CML samples, respectively (Appendix A). We acknowledge the limitation of this trophic annotation methodology and address it further in the Discussion section.

### 2.5. Trophic Index and Trophic Composition

The relative prevalence or success of each trophic group at a given site was expressed as the trophic index of each group (TI*_g_*), where *g* represents three trophic groups of mixotroph (M), autotroph (A), and heterotroph (H). Therefore, TI*_g_* of TI_M_, TI_A_, and TI_H_ can be quantified as their relative abundance against total eukaryotic community (T), either in cell or gene abundances (expressed as M/T, A/T, and H/T). The three combined indices of TI_M_, TI_A_, and TI_H_ represented the complete results for trophic mode compositions at a given site.

### 2.6. Statistical Analysis

The surface dataset was used for redundancy analysis (RDA) to elucidate the overall distribution pattern of trophic groups among 104 stations, and potential relationships with environmental variables. All 29 filtered contextual variables were considered for the analysis, with missing values for each variable replaced with means and then log-transformed for standardization prior to analysis. Hellinger transformation was applied to species abundances and an Escoufier selection with a threshold of 0.9 was applied to the trophic species. Environmental variables that passed both a significance test and a collinearity test were presented on the RDA plot. The RDA was carried out in R using the vegan package, following the script outlined by Faure et al. (2019) [30]. Principal component analysis (PCA) was applied to extract the principal components (PC) that explained the environmental gradients from both surface (SUR) and SUR&CML datasets, using R package pcaMethods [39]. After extracting the main variable components, TI*_g_* indices were plotted over two axes of PC1 and PC2 with a generalized additive model. Stepwise regressions between all filtered variables and TI_M_ were conducted to retrieve important variables explaining mixotrophic prevalence, and linear regressions between single variables and TI*_g_* were carried out to provide additional correlation results. Analysis of variance (ANOVA) was conducted for the culture experiments to reveal factorial/treatment impacts on growth rates and grazing rates among various mixotrophic isolates. 

### 2.7. Culture Experiments

Methods for isolating and cultivating mixotrophs from the North Pacific Subtropical Gyre were described in Li et al. (2021; 2022) [5,40]. Briefly, mixotrophic eukaryotes were isolated from the euphotic zone at Station ALOHA (22°45 ’N, 158°00 ’W), by enriching seawater samples in nitrogen-depleted Keller medium (K-N medium) that was supplemented with bacterial prey. Six identified isolates belonging to Dictyochophyceae, Prymnesiophyceae, and Chrysophyceae were selected for conducting growth experiments. These strains were phylogenetically close to the *Tara* Oceans lineages that were among the top 15 abundant lineages. Prior to experiments, all isolates were acclimated for 1–2 months (duplicated over 10 generations) under corresponding treatment conditions, i.e., low nutrients (K-N medium) and high light, low nutrients and low light, high nutrients (K medium) and high light, and high nutrients and low light. The high and low light treatments were receiving ~100 and ~10 µM photons m^−2^ s^−1^ irradiance on a light/dark cycle (12/12 h). All treatments were supplemented with autotrophic bacterial prey of *Prochlorococcus* (strain MIT9301) at final concentrations of ~2 × 10^6^ cells mL^−1^, which represents the second most abundant prokaryotic population at station ALOHA. Prey cells were centrifuged with a rotation speed of 2000–3000× *g* for 3–5 min and gently resuspended for enrichment. Controls without added prey or grazers were also set up at each condition for comparison purposes. Cell concentrations of prey and grazers were measured every 12–24 h by flow cytometry, and ingestion evidences were occasionally obtained via microscopic observation. 

Based on our previous study on grazing functional responses, most isolates have saturated ingestion rates when the prey concentration is higher than 10^6^ cells mL^−1^. Meanwhile, clearance rates (ingestion rates divided by prey concentration) can roughly present the maximal clearance rates with the final prey concentration used in these experiments. Therefore, we used clearance rates instead of ingestion rates [1,41] to represent grazing performance. Ingestion rates (prey grazer^−1^ h^−1^) for each grazer were calculated as the amount of prey removed between sampling time t + 1 and t, divided by averaged grazer concentrations over the same time period. Clearance rates were calculated by dividing the ingestion rate by averaged prey concentration over the same interval. Body volume-specific clearance rates (body volume grazer^−1^ h^−1^) were calculated by dividing the clearance rate by the cellular biovolume (µm^3^) of each grazer (measured under microscopy) [42]. Clearance rates reported in the results were averaged over all sampling intervals, prior to grazers reaching the stationary phase. The growth rates presented in the results were maximal values during exponential growth phase (slope of Ln transformed cell abundance versus time).

## 3. Results

### 3.1. Oceanographic Characteristics across the Tara Oceans Sampling Regions

The 104 surface samples encompassed eight ocean habitats on a large spatial scale (Figure 1a). Geographically, they spanned roughly four latitudinal regions, from low (0–20°) to low–medium (20–30°), to medium (30–40°), and medium–high latitudes (40–60°). Moving from low to high latitudinal regions, overall PAR decreased from 25.4 to 14.7 mol photons m^−2^ d^−1^, nutrients, e.g., NO_3_^−^&NO_2_^−^, increased from 0.03 to 17.3 µmol L^−1^, and heterotrophic bacterial abundances were relatively stable, ranging between 3.9 and 5.3 × 10^5^ cells mL^−1^ (Table 1; Figure 1b). The globally averaged means were 21.3 mol photons m^−2^ d^−1^ of PAR, 0.14 µmol L^−1^ of NO_3_^−^&NO_2_^−^, and 4.5 × 10^5^ cells mL^−1^ of heterotrophic bacteria, demonstrating most sampling sites were oligotrophic environments. Smaller regional variations were also observed. The higher nutrient concentrations in low-latitude regions were possibly due to the Peruvian coastal and eastern equatorial upwelling, while lower PAR values in medium-latitude areas were because the Mediterranean Sea was sampled during winter.

### 3.2. Relative Abundances and Taxonomic Compositions of Three Trophic Modes 

The trophic index of TI_M,_ TI_A_, and TI_H_ varied from 0.23 to 0.34, from 0.10 to 0.32, and from 0.43 to 0.51 in cell abundance, and from 0.29 to 0.45, from 0.08 to 0.24, and from 0.38 to 0.46 in gene abundance, respectively (Table 1; Figure 1c). TI_M_ values in cell abundance were overall somewhat lower than for gene abundance. Nevertheless, the highest TI_M_ median values were both retrieved from low–medium latitudes, and the lowest values were from low latitudes (cell abundances) and medium–high latitudes (gene abundances), respectively. In comparison, high and low TI_A_ median values were from low and medium–high latitudes and low–medium latitudes (both cell and gene abundances). TI_H_ did not change as much across latitudinal gradients, but the highest median values in cell and gene abundances were shown in low–medium and medium–high latitudes, respectively.

Before the gene and cell abundances correction, mixotrophs were mostly composed of lineages from Dinoflagellata, Haptophyta, and Ochrophyta. After the correction for cell abundances, the dominant mixotrophs were retrieved from Haptophyta and Ochrophyta (Figure 1d). Proportions of dinoflagellates decreased in heterotrophic communities after gene–cell conversion, and the other three most abundant heterotrophs belonged to the phyla Opalozoa, Discoba, and Sagenista, both in cell and gene abundances. Differences in autotrophic compositions on the phyla level in cell and gene abundances were insignificant, which were both dominated by Chlorophyta and Ochrophyta. Redundancy analysis suggested that stations from the same ocean or latitudinal regions tended to cluster together (Figure 1e; Appendix A). Certain species were more abundant in lower latitudes, such as Chrysophyceae_Clade G_sp. of Ochrophyta, *Chrysochromulina* sp. and Haptophyta_HAP2_XXX_sp. of Haptophyta, and *Karlodinium veneficum* of Dinoflagellata for mixotrophs, MOCH.2_XXX sp. and Chlorophyta_XXXX_sp. of Chlorophyta for autotrophs, and MAST.4C_XX sp. and Telonemia Goup2_X sp. for heterotrophs. Those taxa were positively affiliated with PAR and CML, and negatively correlated with nutrients (NO_3_^−^&NO_2_^−^ and NO_2_^−^) and other variables located on the right side of the RDA plot. Other species, including *Geminigera cryophila* of Cryptophyta and *Triparma mediterranean* of Ochrophyta for mixotrophs and Telonemia Group1 for heterotrophs, were abundant at higher latitudes and positively correlated with oxygen and sunshine duration. Additionally, *Micromonas* A2 and Bacillariophyta_XXX sp. for autotrophs were associated with high MLD, POC, and NO_2_^−^, while Cryptomonadales_XX_sp. for mixotrophs and NPK2 lineage_X_sp. for heterotrophs were affiliated with high abundances of *Prochlorococcus*.

### 3.3. Nutritional Variables and the Prevalence of Three Trophic Modes

Among the 29 selected predictors, the availability of nutrients (including NO_3_^−^&NO_2_^−^ and silicate), PAR, and temperature were the top variables revealed by principal component analysis for explaining the environmental conditions of surface (SUR) and surface and chlorophyll maxima layer (SUR&CML) samples (Table 2; Figure 2 and Figure 3). Specifically, both principal component axes (PC1 and PC2) for the SUR&CML samples were mostly explained by PAR and nutrients (loading coefficients varied from −0.42 to 0.17; R^2^ of 0.30). When only considering SUR samples, PC1 axis mostly consisted of nutrients and temperature (R^2^ of 0.35), while the PC2 axis (R^2^ of 0.24) was mostly explained by MLD, CML, and PAR. 

The fitted smoothers of TI*_g_* demonstrating the most substantial responses were from mixotrophs to changing principal components. Specifically, there was a decreasing trend of TI_M_ along PC1 for both datasets, indicating the decreasing prevalence of mixotrophs with increasing nutrients for SUR (Figure 2a), as well as decreasing PAR and increasing nutrients for SUR&CML (Figure 2d). TI_M_ did not demonstrate clear changes with the PC2 axis from SUR samples, so results were not shown. It is noteworthy that the slope for decreasing TI_M_, when nutrients were increasing and PAR was decreasing, was the steepest among all plots (Figure 2d). For the PC2 axis of the SUR&CML dataset, TI_M_ presented a unimodal shape with changing nutrients and PAR conditions, with maximal values obtained around zero PC2 scores (Figure 2g). These results may suggest a more complex non-linear relationship between TI_M_ and co-varying nutrients and light conditions. TI_A_ overall increased with increasing nutrients for the SUR dataset (Figure 2b) and with increasing nutrients and decreasing PAR for the SUR&CML dataset (Figure 2e). When nutrients and PAR changed in the same direction (PC2 for the SUR&CML dataset), TI_A_ did not show clear variations (Figure 2h). TI_H_, overall, varied much less with changing conditions compared to TI_M_ and TI_A_ (Figure 2c,f,i), among which a weak unimodal shape was observed with PC1 from the SUR dataset (Figure 2c). For the gene abundance-denoted results, similar trends were found for TI*_g_* variations over the principal components but with different slope shapes (Figure 3). Particularly, a steeper decreasing slope was presented for TI_M_ on the PC1 axis of the SUR samples and a steeper increasing slope (before the peak) was seen on PC2 for the SUR&CML samples (Figure 3a,g). The weak increasing trend of TI_H_ over PC1 axis of SUR samples disappeared (Figure 3c), and a somewhat stronger increase in TI_H_ was demonstrated along both axes of PC1 and PC2 for the SUR&CML dataset (Figure 3f,i).

Consistent with the PCA results, clear differences were observed for TI*_g_* between SUR and CML samples (from 61 stations, we have both SUR and CML samples), accompanied with distinct resource availability of nutrients, light, and bacterial prey (Figure 4). Significantly higher TI_M_ values were seen in surface samples where PAR and bacteria were higher and N (NO_3_^−^&NO_2_^−^) was lower (median 0.29 vs. 0.18, *p* < 0.001, df = 60). In comparison, TI_A_ was slightly higher in CML compared to surface samples, but this was not statistically significant (median 0.30 vs. 0.29, *p* > 0.05, df = 60), and TI_H_ values were significantly higher in CML than surface samples (median 0.52 vs. 0.47, *p* < 0.05, df = 60) (Figure 4a,b). Linear regressions derived from all samples suggested a consistently positive relationship for TI_M_ with PAR and negative correlations with nutrients (NO_3_^−^&NO_2_^−^), from both the SUR (122 samples; Figure 4c) and the SUR&CML datasets (165 samples; Figure 4d). TI_A_ demonstrated a significantly positive and negative correlation with nutrients and PAR, respectively, whereas neither was significantly associated with TI_H_ (Appendix A). For heterotrophic bacteria, a positive but weaker relationship (*p* = 0.02) was found between TI_M_ and the SUR&CML dataset, but not with the SUR dataset (Figure 4c,d). In contrast, both datasets demonstrated strong negative associations between heterotrophic bacteria and TI_H_ values, and a weaker but positive correlation between heterotrophic bacteria and TI_A_ in the surface (*p* < 0.05) (Appendix A). Other single variables showing significant regressions with TI_M_ were all shown in Appendix A. 

### 3.4. Mixotrophic Growth with Varying Nutritional Resource

The most abundant mixotrophs identified from the *Tara* Oceans dataset belonged to the classes of Prymnesiophyceae, Dictyochophyceae, Chrysophyceae, Dinophyceae, and Cryptophyceae (Figure 5a). Mixotrophic strains that we isolated from the North Pacific had been previously shown to be efficient grazers of heterotrophic bacteria and cyanobacteria (including *Prochlorococcus* and *Synechococcus*), and they were closely affiliated to those abundant *Tara* Oceans lineages [5,40]. Six strains selected for the experimental study belonged to *Chrysochromulina*, Prymnesiophyceae_XXX of Prymnesiophyceae, Clade-H_X of Chrysophyceae, as well as *Florenciella*, Dictyochales_X, and Florenciellales_X of Dictyochophyceae. Although the lineage from Florenciellales_X was not shown in Figure 5a, it ranked as the 26th most abundant lineage across all stations. The relative abundances of those isolates retrieved from surface samples were all higher than from CML, except for Dictyochales_X (Figure 5b), presenting similar patterns as the entire mixotrophic community (TI_M_ results in Figure 4a). Given the consistent and substantial responses of TI_M_ to nutrients and PAR derived from field observations, we designed multifactorial bioassay experiments to assess the impacts of light and nutrient (inorganic and organic nutrients) availability on the growth of mixotrophs (Figure 5c). 

Distinct growth curve shapes and final abundances were presented among different treatments for the six isolates (Figure 6a), accompanied by different disappearance rates of prey (Figure 6b). Prey in controls (without mixotrophic grazers) all presented slower disappearance rates compared to cultures with grazers, but to different levels among strains. Grazers in the low nutrient control (without prey) grew to much lower concentrations, i.e., all <1 × 10^3^ cells mL^−1^, compared to 10^4^–10^5^ cells mL^−1^ when prey was added. Shown as exponential growth rates (bars in Figure 6c), most isolates presented the highest values when grown under high amounts of light, either when receiving high nutrients or sufficient prey (low nutrients). Lower growth rates were observed with low-light treatments (marked with black arrows in Figure 6c) for strain 4110 from *Chrysochromulina*, 4150 from Prymnesiophyceae_XXX, and 3021 from Florenciellales_X, i.e., those categorized as trophic type 1. Growth of one isolate from *Florenciella* (strain 3010) and one isolate from Chrysophyceae Clade H_X (strain 3501) did not show significant impacts by light and, thus, were classified as trophic type 2. One additional strain 3050 from Dictyochales_X represented slightly different characters from both Type 1 and Type 2 and were thus categorized as Type 1/2. Specifically, they performed better under low light than Type 1 strains when receiving high nutrients, but not as good as Type 2 strains with low light and low nutrients. More experiments needs to be done to verify its trophic traits. For grazing rates, most strains showed the highest clearance rates when receiving low levels of nutrients and high amounts of light (marked with blue arrows in Figure 6d). Only two strains belonging to Type 2 were capable of grazing much faster when receiving low light levels and high nutrient levels (marked with red arrows in Figure 6d). 

The ANOVA results revealed a significant effect on growth by light, i.e., to increase growth rates with increasing light, either alone, or together with nutrients and species. However, (inorganic) nutrient availability did not show significant impact on growth rates, as cultures supplemented with bacterial prey can sufficiently compensate for nutrient deficiency. Less significant effects were observed for clearance rates, as strains responded differently to changing light and nutrient conditions, indicated by the high *p* (>0.05) values from both single and multi-factorial ANOVAs (Table 3).

## 4. Discussion

In this study, we used large-scale field surveys to retrieve prominent variables that possibly affect the trophic mode structure of eukaryotes on community levels, and across large spatial scales, with a particular focus on three nutritional resources. We then conducted finely controlled lab experiments using isolates to investigate how mixotrophic growth responds to changing nutritional resources, in terms of both forms (inorganic nutrients and organic nutrients) and availability. While acknowledging species-specific variations, both field and experimental results demonstrated a close relationship between mixotrophic prevalence and nutritional resource among inorganic nutrients, light, and prey, which possibly introduced distinct niche preferences for mixotrophs versus autotrophic and heterotrophic eukaryotes in the ocean.

### 4.1. Mixotrophic Growth Responding to Changing Nutritional Resources 

Suggested by our experimental results, mixotrophy can function as an efficient strategy for compensating for inorganic nutrient deficiency for most strains through the ingestion of bacterial prey, but not necessarily for supplementing light limitation (Figure 6c). Among the six studied cultures, two of them were efficient with compensating light/energy deficiency via heterotrophic nutrition (grazing), but not the others. These results indicated that mixotrophic controlling mechanisms for metabolic partitioning between autotrophy and heterotrophy were species-specific. The Type 1 isolates that were light-dependent possibly align with the ‘primarily photosynthesis and phagocytosis for nutrients’ functional group proposed by Jones (1997) and Stoecker (1998) [43,44]. The growth rates of Type 2 isolates seemed to be light- or nutrients- independent, and likely belong to the ‘phagocytose when light is limiting’ or ‘perfectly balanced mixotrophy’ functional group. The Type 1/2 strain showed varying responses to light when nutrients were high or low, and are thus seemingly to be in between or different from the other two trophic types. It is also possible that the culture under high nutrient and low light were not well acclimated for grazing, and more experiments need to be conducted for further verification. Based on proposed resource allocation theories, those light-dependent mixotrophs (Type 1) probably rely on photosynthesis for gaining most energy and organic carbon [12,13,14]. In comparison, the light-independent group performed better with low light availability (Type 2), suggesting a higher flexibility governing autotrophic and heterotrophic growth in response to the changing conditions. 

### 4.2. Relationship between Mixotrophy and Nutritional Resources on Large Spatial Scale

Our field analysis also indicated that availability of three nutritional resources could affect the relative success among three trophic strategies on community level, and across large spatial scales. Specifically, higher-nutrient and lower-light environments could possibly promote the relative abundances of autotrophs, whereas mixotrophic community overall proliferated from low-nutrient and high-light environments. This positive light–mixotrophy relationship in the field can be partially explained by the possibility that the light-dependent Type 1 mixotrophs were dominant across the investigated ocean regions. On the other hands, Rothhaupt (1996) and Edwards et al. (2019; 2022) suggested that, during the resource competition process, mixotrophs could outcompete heterotrophs, because they can use extra energy gained from photosynthesis which enables them to suppress bacterial prey to a very low level (below the threshold of heterotrophic grazers) [8,11,45]. Both mechanisms could be true, and it is worthy to conduct further studies to investigate the fundamental mechanisms related to metabolic partitioning and resource allocation for mixotrophic strategy.

Heterotrophic bacteria maintained relatively consistent abundances in the sampled oceans and was not selected as main explanatory factor for TI*_g_* by RDA or PCA. Nevertheless, they demonstrated one of the predictors explaining TI_M_, based on a stepwise regression model (Appendix A). When treated as a single variable, a strong negative correlation was found between TI_H_ and heterotrophic bacteria. Comparatively, the correlation was positive for TI_A_ and less significant for TI_M_. This interesting shift likely suggested distinct interactions between bacteria and three trophic groups. Heterotrophic bacteria benefit from the dissolved organic carbon (DOC) excreted by autotrophs and recycle nutrients for autotrophic growth, forming a positive relationship, whereas, for heterotrophic eukaryotes, this negative association could be attributed to the high grazing pressure they exert on heterotrophic bacteria [46,47]. The weaker but positive correlation between heterotrophic bacteria and TI_M_ could be explained by the more complex relations among bacteria (prey and decomposer) and mixotrophs (grazer and DOC provider), as well as the lower grazing capability/pressure from mixotrophic grazers than heterotrophs [13,23,38,48].

### 4.3. Environmental Variables Affecting the Prevalence of Trophic Strategies

Other oceanographic conditions such as CML, nitracline depths, and MLD also seemed to affect mixotrophic prevalence, though they are physical processes that are more or less associated with nutrient and light conditions. For example, deeper nitraclines can potentially suppress nutrient supply in the surface and, therefore, increase TI_M_ values_._ Similarly, deeper chlorophyll maxima could be indicative of a higher light availability and lower nutrient availability in the photic depths, thereby enhancing TI_M_. In contrast, deeper mixed layer depths (stronger mixing) could reduce light availability and increase nutrient concentration, which both weaken TI_M_. The overall positive association between CML and nitracline depths and TI_M_, and the negative correlation between MLD and TI_M_, are in line with their overall relationships with nutrients and light. 

Previous field studies suggested that, besides nutrients [21,49] and light [50,51,52], variables such as temperature [53] and carbon composition (colored dissolved organic matters and CO_2_) [20] were also likely important for affecting mixotrophic prevalence. These variables were also included in our analysis and yielded varying relationships with TI*_g_*. These differences could possibly be due to the different environmental gradients covered among our studies (various open oceans) versus others (North Pacific, and boreal Canadian lakes). The stepwise regression model suggested that PAR, nutrients, density, PIC, CML, heterotrophic bacteria, and *Prochlorococcus* were the top predictors explaining TI_M_ for the *Tara* Oceans dataset (Appendix A). 

Indeed, it is not always straightforward to delineate relationships between a single environmental variable and trophic mode prevalence. Results from our statistical analysis instead provide insights into the possible correlations between them, and unnecessarily the causational relationships. Future studies focusing on smaller regions, with in situ experiments targeting active trophic behaviors, will provide additional knowledge for interpreting the relationships between trophic mode and the environment. 

### 4.4. Methodology Caveats 

One potential caveat in our methodology is that mixotrophs were annotated based on the published literature at a species or higher taxonomic rank. These trophic annotations did not necessarily align with in situ metabolic activities. Nevertheless, the potential or historically evolved capability of phago-mixotrophy was sufficient to delineate their ecological niche, which was the focus of this study. We also recruited longer-term averaged values for some of the key variables (e.g., 8-day and 30-day PAR), as well as a variety of parameters (e.g., NO_3_^−^, NO_2_^−^, PO_4_^3−^ and SiO_4_^−^ for nutrients), to reduce the possibility of retrieving false relationships. A lack of trophic evidence for some of the species/lineages could, on the other hand, cause false annotation, and variations against other studies. For example, we annotated *Micromonas*, a member of Chlorophyta, as autotrophs, based on the best knowledge we have [43,54]. However, others have argued for possible phagotrophic evidence in *Micromonas polaris* [55]. We also recognize that the gene and cell abundance correction factors could be inaccurate for certain groups, as we did not have first-hand cellular abundance data from in situ samples. Nevertheless, these calibrated abundance data could improve accuracy for some heavily over-estimated groups, such as dinoflagellates, and for under-estimated small flagellates.

Combining various approaches including experimental study, field observation, model simulation, and bioinformatics tools can be a powerful solution in terms of retrieving the most reliable data for future mixotrophy research [3,56].

## 5. Conclusions

This study provides the first *Tara* Oceans analysis for niche partitioning among three trophic populations of mixotrophs, autotrophs, and heterotrophs of the eukaryotic community. Our results suggested that, either on a community level or a species level, phagotrophic mixotrophy is a trophic strategy that marine eukaryotes have evolved to survive nutrient impoverishment and/or light limitation, deepening our understanding about the mechanisms controlling their prevalence in the ocean.

## Figures and Tables

**Figure 1 microorganisms-12-00750-f001:**
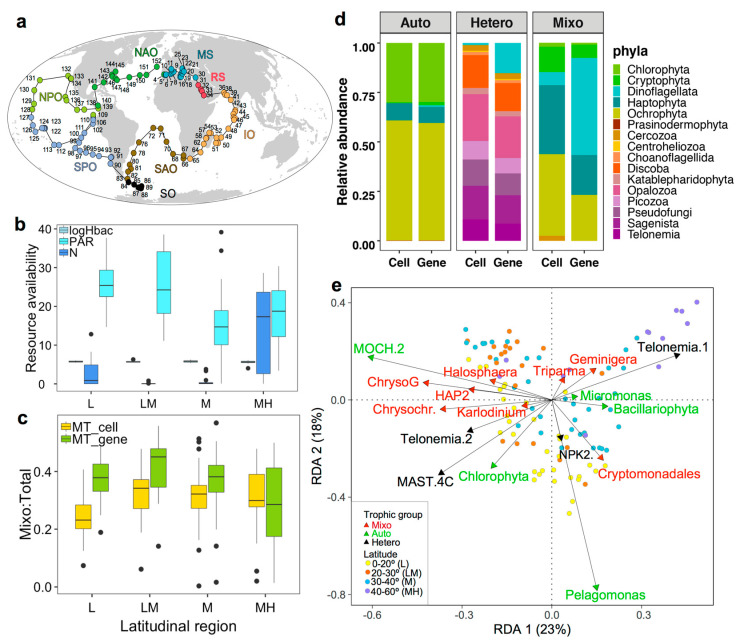
Map of study sites (**a**), averaged median values of inorganic nutrients (NO_3_^−^&NO_2_^−^; N for abbreviation), light (PAR) and heterotrophic bacteria (Hbac) (**b**), and mixotrophic prevalence index TI_M_ (Mixo/Total) derived from 18S rRNA gene (MT_gene) and corrected cell abundances (MT_cell) (**c**) at different latitudinal regions. (**d**) presents the composition of autotrophic (Auto), mixotrophic (Mixo), and heterotrophic (Hetero) eukaryotes on a phyla level, derived from 18S rRNA gene and corrected cell abundances. (**e**) is a redundancy analysis plot showing the distribution of trophic groups on a species level. Oceanic region abbreviations in panel (**a**) are, NAO: North Atlantic Ocean; MS: Mediterranean Sea; RS: Red Sea; IO: Indian Ocean; SAO: South Atlantic Ocean; SO: Southern Ocean; SPO: South Pacific Ocean; NPO: North Pacific Ocean. Latitudinal region names and definitions in (**b**,**c**) are shown in the legend of (**e**), and black dots were potential outliers from individual samples. *Tara* Oceans lineages in (**e**) were shown with genera name, and full names of species can be found in Appendix A and Appendix A.

**Figure 2 microorganisms-12-00750-f002:**
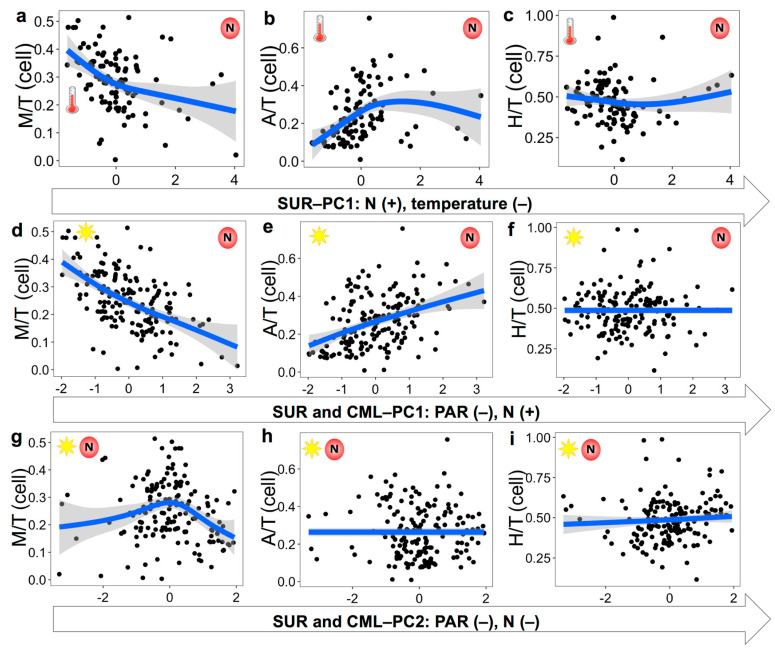
Variation patterns of TI_A/M/H_ in corrected cell abundances along PC1 axis for surface (SUR) dataset (**a**–**c**) and PC1 and PC2 axes for surface and chlorophyll maximal layer (SUR&CML) dataset (**d**–**i**). Cartoon symbols of nutrients (NO_3_^−^&NO_2_^−^) (letter N with a red circle), PAR (the sun), and temperature (a thermometer) were marked on each panel, and the location of them indicates higher values ends. Each black dot represents observed value from two datasets. Blue lines and shaded bands were fitted to a generalized additive model smoother and with 95% confidence intervals.

**Figure 3 microorganisms-12-00750-f003:**
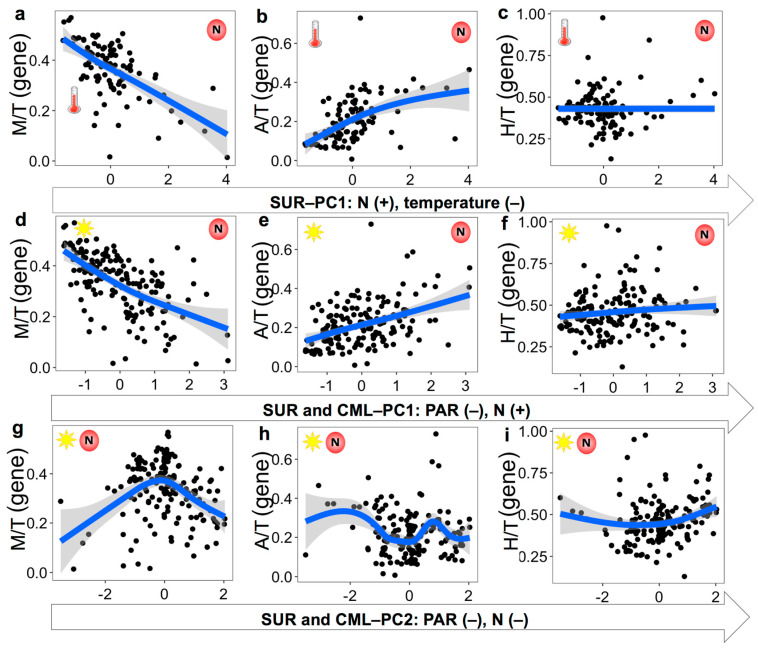
Variation patterns of TI_A/M/H_ in 18S rRNA gene abundances along the PC1 axis for the surface (SUR) dataset (**a**–**c**) and PC1 and PC2 axes for the surface and chlorophyll maximal layer (SUR&CML) dataset (**d**–**i**). Cartoon symbols of nutrients (NO_3_^−^&NO_2_^−^) (a letter N with a red circle), PAR (the sun), and temperature (a thermometer) were marked on each panel, and the location of them indicates higher values ends. Each black dot represents observed value from two datasets. Blue lines and shaded bands were fitted to the generalized additive model smoother with 95% confidence intervals.

**Figure 4 microorganisms-12-00750-f004:**
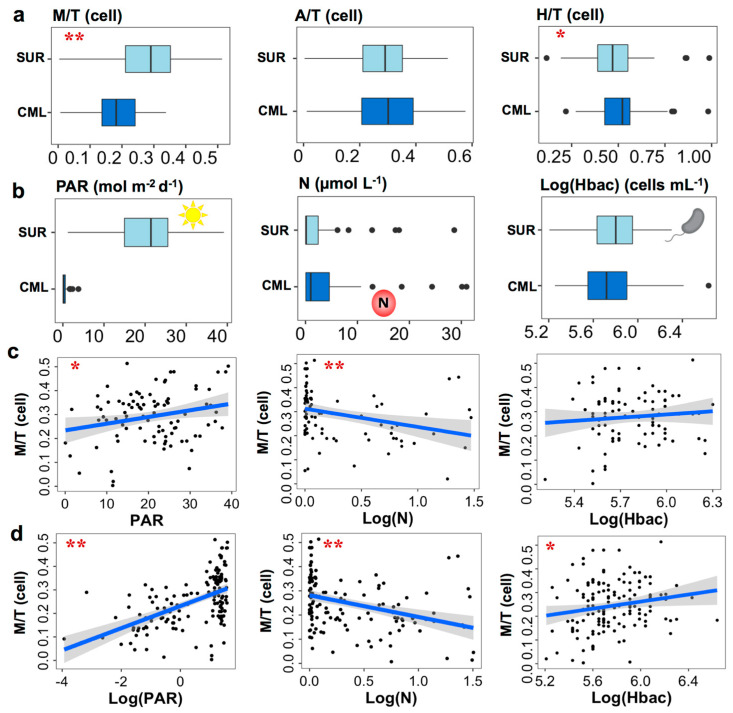
Comparison of TI*_g_* (**a**) and nutritional resource availability (**b**) between surface (SUR) and chlorophyll maximum layer (CML). (**c**,**d**) demonstrate linear regressions between TI_M_ and three resource variables of PAR, NO_3_^−^&NO_2_^−^ (N), and heterotrophic bacteria (Hbac) with the SUR dataset (**c**) and the SUR&CML dataset (**d**), respectively. Variables in (**c**,**d**) were sometimes Log-transformed for the best representation purpose, and each black dot represents observed value from two datasets. Significance codes were shown using red stars (one star if 0.001 < *p* < 0.05 and two stars if *p* < 0.001), and shaded bands are pointwise 95% confidence intervals on the fitted linear regression (blue lines). The cartoon symbols of the sun, letter N with red circles, and bacterial cell shape represents PAR, NO_3_^−^&NO_2_^−^, and Hbac concentrations, respectively.

**Figure 5 microorganisms-12-00750-f005:**
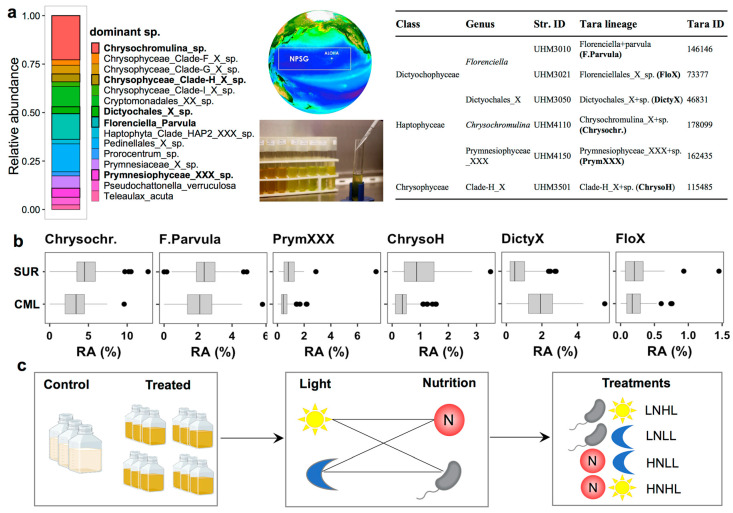
Relative abundances of the top 15 most abundant *Tara* Oceans mixotrophic lineages (in gene converted cell abundances), and phylogenetically related strains isolated from station ALOHA in the North Pacific Subtropical Gyre (**a**). Comparison of relative abundances in surface (SUR) and chlorophyll maximum layer (CML) (**b**), and an illustrative flow chart showing how different treatments were set up for the growth experiments (**c**). Cartoon symbols of the sun, the moon, bacterial shape and letter N with circles in (**c**) stand for treatment of high light, bacterial prey as the only primary nutrition (low inorganic nutrients), low light and high inorganic nitrogen levels, respectively. Images resources of two middle pictures in (**a**) are the Hawaiian Ocean Time Series crew and Dr. C.R. Schvarcz (was previously affiliated with the University of Hawaii).

**Figure 6 microorganisms-12-00750-f006:**
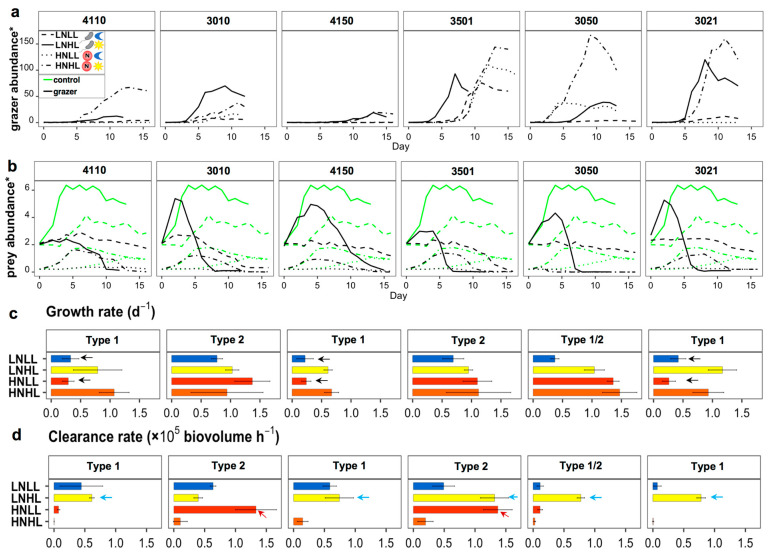
(**a**) presents growth curves of six mixotrophic isolates grown under high light (HL) and low light (LL), as well as high inorganic nitrogen (HN) and low inorganic nitrogen (LN) conditions. All treatments were supplemented with *Prochlorococcus* prey. (**b**) presents removal of prey in all cultures (ID table in Figure 5a); green lines indicate prey control without grazers and black lines were prey with grazers. (**c**,**d**) demonstrate estimated growth rates during exponential phase and biovolume clearance rates for each strain. Errors bars in (**c**,**d**) denote standard deviation among replicates for each treatment, which were not shown in (**a**,**b**) for higher readability. ***** For grazer concentrations in HNHL, they were divided by 50,000 cells mL^−1^ for strain 4110 and 5000 cells mL^−1^ for other strains, and 1000 cells mL^−1^ for all the other treatments. * Prey abundances in all HN treatments were divided by 10^7^ cells mL^−1^, and all LN were divided by 10^6^ cells mL^−1^. Arrows in (**c**,**d**) indicate treatments that caused significantly lower growth rates (**c**), and significantly higher grazing rates (**d**).

**Table 1 microorganisms-12-00750-t001:** Oceanographic characters and trophic index of surface water samples. The abbreviations PAR, NO_3_^−^&NO_2_^−^, and Hbac stand for photosynthetically available radiation, dissolved nitrate and nitrite (NO_3_^−^&NO_2_^−^), and heterotrophic bacteria, respectively. M/T, M/A, and M/H denote rations of either corrected cell abundance (Cell) or 18S rRNA gene abundance (Gene) among mixotrophs (M), autotrophs (A), heterotrophs (H), and total eukaryotes (T).

Latitude	Oceanic Region ^†^	Station	Temp (°C)	Salinity	PAR (mol m^−2^ d^−1^)	NO_3_^−^&NO_2_^−^ (µmol L^−1^)	Hbac (Cells mL^−1^)	M/T (Cell/Gene)	A/T (Cell/Gene)	H/T (Cell/Gene)
0–20°	SPO, IO, NPO, SAO, NAO	36	26.6	35.1	25.4	0.89	4.9 × 10^5^	0.23/ 0.38	0.32/ 0.24	0.43/ 0.38
20–30°	IO, NPO, SPO, SAO, RS	17	24.7	36.2	24.3	0.03	3.9 × 10^5^	0.34/ 0.45	0.10/ 0.08	0.51/ 0.44
30–40°	MS, SAO, NAO, SO	37	19.5	36.4	14.7	0.13	5.3 × 10^5^	0.32/ 0.38	0.19/ 0.16	0.48/ 0.43
40–60°	SO, SAO, MS	14	7.1	34.3	18.8	17.3	4.5 × 10^5^	0.30/ 0.29	0.18/ 0.22	0.47/ 0.46
Global	SPO, IO, NPO, SAO, NAO, SO, RS, MS	104	23.8	35.4	21.3	0.14	4.5 × 10^5^	0.29/ 0.38	0.21/ 0.19	0.47/ 0.42

^†^ Oceanic regions follow the *Tara* Oceans standards. NAO: North Atlantic Ocean; MS: Mediterranean Sea; RS: Red Sea; IO: Indian Ocean; SAO: South Atlantic Ocean; SO: Southern Ocean; SPO: South Pacific Ocean; NPO: North Pacific Ocean.

**Table 2 microorganisms-12-00750-t002:** Major environmental variables retrieved by principal component analysis for surface (SUR) and surface and chlorophyll maximal layer (SUR&CML) datasets. Top variables from each principal component axis (PC1 and PC2) for both datasets were shown with their loading coefficients (marked in bold). Symbols of ‘−’ explain the changing variable values decrease from left to right on the axes.

	SUR		SUR&CML		
	PC1	PC2	PC1	PC2	Sum ^†^
PAR	−0.08	**−0.14**	**−0.42**	**−0.35**	0.99
NO_3_^−^&NO_2_^−^	**0.31**	**−0.14**	**0.23**	**−0.25**	0.93
Silicate	**0.23**	−0.10	**0.17**	**−0.17**	0.67
Temperature	**−0.24**	0.08	−0.14	0.13	0.59
Nitracline depth	−0.20	−0.11	−0.15	0.12	0.58
CML	0.06	**−0.17**	−0.10	0.07	0.40
Mixed layer depth	0.06	**−0.22**	−0.04	−0.06	0.38
R^2^	0.35	0.24	0.30	0.25	

^†^ summed absolute values from PC1 and PC2 for both datasets.

**Table 3 microorganisms-12-00750-t003:** ANOVA of different growth rates and clearance rates among six species cultivated under different light (PAR) and nutrient (N) conditions (results shown in Figure 6). Individual and interactive effects of light, nutrients, and species were analyzed via single factor and multi-factor ANOVAs, respectively.

	Growth Rates					Clearance Rates				
	Single Factor			Multi-Factors		Single Factor			Multi-Factors	
	PAR	N	Sp.	PA and N	PAR and N and Sp.	PAR	N	Sp.	PAR and N	PAR and N and Sp.
*p* ^†^	*	>0.05	>0.05	*	*	>0.05	>0.05	>0.05	*	0.13
F ^‡^	6.5	1.7	2.1	5.1	3.6	0.00	2.98	1.35	3.96	2.02
Dfn ^§^	1	1	5	3	5	1	1	5	3	5
Dfd ^‖^	20	20	17	18	18	20	20	17	18	18
Effect ^¶^	+	+	/	/	/	+	–	/	/	/

^†^ Significance level of ANOVA, ‘*’ denotes *p* < 0.05. ^‡^ Variation between sample means/variation within the samples. ^§^ Degrees of freedom in the numerator (Dfn), equals to the groups (of factors). ^‖^ Degrees of freedom in the denominator (Dfd), equals to the total number of subjects in the experiment—the groups (of factors). ^¶^ Symbols of + and – indicate positive and negative impacts from increasing light and nutrient availability, and / denotes “analysis unavailable”. Only factors with *p* values marked with * in the first row had a significant impact.

## Data Availability

All datasets used in this study can be downloaded from the Appendix A. Original metabarcode and environmental context data were publicly deposited by the *Tara* Oceans Project, under Pangaea website (https://doi.pangaea.de/10.1594/PANGAEA.840721 and European Nucleotide Archive (project accession #PRJEB6610).

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
