# Peer review of "Prevalence and Preferred Niche of Small Eukaryotes with Mixotrophic Potentials in the Global Ocean"

_microorganisms, 2024, doi:10.3390/microorganisms12040750_

Round 1
Reviewer 1 Report
Comments and Suggestions for Authors
The article is, from the point of view of molecular biology very well presented and it contains very interesting treatment of molecular data. I would recommend it for a quick publication. However...
Methodology:
133 DCM is in use as the Deep chlorophyll maximum, which is confusing in your manuscript. Or is it?
152 To omit ciliates and rhizarians from the analysis could be reasonable, it is your decision. On the other hand, they are principal predators of the analysed size fraction. It is a theme for discussion. I am not native but to me, it does not sound well formulated in English.
166 Regarding a eukaryotic assemblage (I am an orthodox - community is plankton), I would prefer not to enter into details: "small flagellates", which include diatoms and phytoflagellates... it sounds rare to me. Yes, sex cells of diatoms are flagellated and some chlorophytes are flagellated, too, but the whole formulation could be changed.
229 Prochlorococcus is a picocyanobacterium thus "gentle centrifugation" was probably using high x g.
229 cyanobacteria are bacteria but ecologically, as a prey, may be very different. In addition to the particle size, their content of photosynthesising- and accessory pigments make them different from heterotrophic bacteria and they might be ingested depending on environmental variables, e.g., light or DO. I would prefer to mention them separately, not as "included"; I accept, it is subjective.
235 The equations to be used in microbial predation were formulated by Fenchel and/or by the Sherrs and the credits should be put. Selectivity is reviewed and discussed, e.g., by Montagnes et al. 2008. The recent overview is by Weisse et al. 2016 if you want to use a fresh article even though it is based on the above-mentioned authors and not all parameters, which you are using are included. Ingestion rate (frequently named uptake rate and in other context grazing rate) is mathematically analysed in Vaque et al. 1994 and you can see that your method is well supported. However, I am lacking a citation. Specific clearance rate or specific clearance is not so frequently used (even though it is an excellent parameter) and who has introduced it to the practice was Fenchel (1980 or 1986)
There is a "behavioural" problem with the grazer, because within microbial ecologists it is frequently used for non-selective feeding organisms - suspension feeders (originally cattle, in the context, e.g., ciliates feeding by filtering). In the case of your phagotrophs, many of them are known bacteria hunters or they are using another type of individual capture of the prey - direct interception. I prefer to use the feeder, which is general. However, my opinion is subjective.
287 Opalozoa and even more Discoba are problematic if you want to take them into account as homogeneous groups. Parasites, phagocyting and photosynthesising species, respirating- anaerobic- and facultatively anaerobic mitochondrion, etc.,make them very complex taxons to be interpreted
From the point of view of scientific publication, there are plenty of incorrect formats:
(i) In the text, the ions must be put with charges and numbers of oxygen as an index (e.g., NO3- ); the same for carbon dioxide, etc. I agree that in the graphs produced by some programmes, it will not be
(ii) Latin names both in the text and the tables must be in italics and the species name must be in lowercase (e.g., Triparma mediterranea)
(iii) The literature is not cited in the same way

Author Response
To the Editor:
−We thank the two reviewers for their insightful comments. The majority of the suggestions, if not all, have been taken into account and incorporated into the revised manuscript (detailed answers were listed in the section below). We have made some additional minor revisions to improve the writing quality and the soundness of conclusions.
Reviewer 1
The article is, from the point of view of molecular biology very well presented and it contains very interesting treatment of molecular data. I would recommend it for a quick publication. However...
−We thank Reviewer 1 for her/his detailed and informative comments. In particular, the suggestions on the technical aspects have aided us in improving the quality and organization of the manuscript. We have addressed each of the reviewer's suggestions and revised the manuscript accordingly. We believe that the new version of the manuscript represents a significant improvement.
Please note that we use color highlights throughout the manuscript to indicate revisions and changes.
133 DCM is in use as the Deep chlorophyll maximum, which is confusing in your manuscript. Or is it?
Answers: Thank you for your suggestion. We agree with your advice. We have made changes accordingly throughout the manuscript. We now use CML as the corrected abbreviation instead to avoid confusion, which denotes the Chlorophyll Maximum Layer, rather than deep or depth of chlorophyll maximum.
152 To omit ciliates and rhizarians from the analysis could be reasonable, it is your decision. On the other hand, they are principal predators of the analysed size fraction. It is a theme for discussion. I am not native but to me, it does not sound well formulated in English.
Answers: This is a very good point. We also debated on whether we should keep ciliates and rhizarians in the datasets. The rationale behind removing them is partially because they did not account for a large proportion for the entire eukaryotic community so the reults would have not changed much. We now added a sentence addressing this issue on Page 5, Line 183-186).
166 Regarding a eukaryotic assemblage (I am an orthodox - community is plankton), I would prefer not to enter into details: "small flagellates", which include diatoms and phytoflagellates... it sounds rare to me. Yes, sex cells of diatoms are flagellated and some chlorophytes are flagellated, too, but the whole formulation could be changed.
Answers: We have changed ‘(other) flagellates’ into ‘other eukaryotes’ throught the manuscript.
229 Prochlorococcus is a picocyanobacterium thus "gentle centrifugation" was probably using high x g.
Answers: Thank you for pointing out this.We have now added detailed information for the centrifuge processes with the rotation speed (Page 7, line 261-262).
229 cyanobacteria are bacteria but ecologically, as a prey, may be very different. In addition to the particle size, their content of photosynthesising- and accessory pigments make them different from heterotrophic bacteria and they might be ingested depending on environmental variables, e.g., light or DO. I would prefer to mention them separately, not as "included"; I accept, it is subjective.
Answers: We agree that autotrophic and heterotrophic bacteria are ecophysiologically different. They are basically one primary producer and one primary consumer/decomposer. As a prey resource for the mixotrophic grazers, on the other hand, these two groups do not show functional differences. As stated in our previous publications (Li et al., 2021; 2022), these mixotrophic eukaryotes isolated from open ocean can remove/feed on both bacterial groups efficiently (cyanobacteria and heterotrophic bacteria). We have paid extra attention to the way we describe them in the new manuscript and made changes accordingly (Page 9, line 376-377; Page 2, line 79-80).
I recommend to put cyanobacteria explicitely, e.g., heterotrophic bacteria and cyanobacteria (including...
Answers: We take the reviewer’s suggestion and made changes for line 376-377.
235 The equations to be used in microbial predation were formulated by Fenchel and/or by the Sherrs and the credits should be put. Selectivity is reviewed and discussed, e.g., by Montagnes et al. 2008. The recent overview is by Weisse et al. 2016 if you want to use a fresh article even though it is based on the above-mentioned authors and not all parameters, which you are using are included. Ingestion rate (frequently named uptake rate and in other context grazing rate) is mathematically analysed in Vaque et al. 1994 and you can see that your method is well supported. However, I am lacking a citation. Specific clearance rate or specific clearance is not so frequently used (even though it is an excellent parameter) and who has introduced it to the practice was Fenchel (1980 or 1986)
Answers: Very good point! We have added the relevant references mentioned and explained the reason why we used clearance rates instead of the standard ingestion rates to show grazing performance (Page 7, line 266-271).
There is a "behavioural" problem with the grazer, because within microbial ecologists it is frequently used for non-selective feeding organisms - suspension feeders (originally cattle, in the context, e.g., ciliates feeding by filtering). In the case of your phagotrophs, many of them are known bacteria hunters or they are using another type of individual capture of the prey - direct interception. I prefer to use the feeder, which is general. However, my opinion is subjective.
Answers: We agree with you. However, in the current study, we study small eukaryotic organisms that are capable of phagocytosis. We do not emphasize on their feeding behavior types, so grazers will be kept as it is. We will keep your suggestion in mind for the future research though.
287 Opalozoa and even more Discoba are problematic if you want to take them into account as homogeneous groups. Parasites, phagocyting and photosynthesising species, respirating- anaerobic- and facultatively anaerobic mitochondrion, etc.,make them very complex taxons to be interpreted.
Answers: We thank you for this valuable comments. Indeed, it is a challenging work to annotate trophic strategy based on taxonomic information. We therefore acknowledge the limitation of this method (Page 12, line 495-510).
From the point of view of scientific publication, there are plenty of incorrect formats: (i) In the text, the ions must be put with charges and numbers of oxygen as an index (e.g., NO3- ); the same for carbon dioxide, etc. I agree that in the graphs produced by some programmes, it will not be
Answers:We have made changes accordingly throughtout the manuscript (highlighted colors).
(ii) Latin names both in the text and the tables must be in italics and the species name must be in lowercase (e.g., Triparma mediterranea)
Answers:We have made changes accordingly throughtout the manuscript (highlighted colors).
(iii) The literature is not cited in the same way
Answers: We have reorganized the format of the references according to the citation style of the MDPI journal.

Reviewer 2 Report
Comments and Suggestions for Authors
Overall, this is an interesting exploration of the prevalence and preferred niche of small eukaryotes with a focus on the role of mixotrophs in the global ocean. The authors merge data from the Tara Oceans eukaryotic 18S rRNA gene dataset and environmental variables to conduct large- scale field analysis, and isolates-based culture experiments to verify growth and nutritional resource relationships for representative taxa. Although the authors made some simplifying assumptions and adjustments in the data to permit a combined analysis of the total data set (which they acknowledge in the text and discussion), the overall results are informative and in many cases comport with prior published results on some of the key relationships that the authors report in their findings. In general, the research illustrates the value of this kind of merging of diverse data sources to provide a richer interpretation of open ocean dynamic processes.
Corrections for the authors to consider:
Line Correction
12 Unicellular eukaryotes that are capable of phago-mixotrophy in the ocean compete for inorganic nutrients and light----
18-20 i---- field analysis, and isolates-based culture experiments to verify growth and nutritional resource relationships for representative taxa. The field observations showed that compared to autotrophic and heterotrophic eukaryotes,----
27 This study revealed a close relationship between---
39-40 --- in fact mixotrophs, phagotrophic mixotrophs particularly, that are capable of both—
50 mixotrophs and autotrophs compete for (inorganic) nutrients---
60 -- this universal mixotrophic trait; that is, photosynthesis and/or--
72 With climate change, evidence has shown that---
86 --- each lineage/species and assess relationship between---
104-105 Bathymetry depths varied from 13 to 5964 km and distance to coast was between 0.9-1404 km.
107 We built an environmental context dataset for further statistical analysis---
139 --- distance to coasts, sampling depths and bathymetry depths were---
157 Eukaryotic communities are known to have---
167-168 ---chloro-phytes, etc.). Finally, a correction factor (C.F.) for each group was calculated---
170 --- other small flagellates, respectively.
172 --- cell abundances by dividing gene numbers with C.F.—
174 --- were also assessed occasionally for comparison purposes.
179 --- each lineage/species (when possible) into three trophic groups,---
182-183 ---- (pigmented protists with no available evidence of phagocytosis) by searching evidence among published databases and references---
184-185 ---- at lineage or species level, their---
189 Can the authors please check this reported dataset umber ---- a final trophic composition dataset with 58,001,584 sequences----
191-192 --- methodology, and address it further—
200 Surface datasets (SUR) were used---
209-210 -- that explained the environmental gradients from both---
212 -- extracted, TIg indices were plotted—
253-254 The higher nutrient concentrations in low latitude regions were possibly due to --
256-257 Sea was sampled in winter. Three trophic indices of TIM,==
AUTHORS please note you have misspelled principal component analysis throughout the manuscript. Your misspelled wording is printed as “principle component analysis –”
360 --- between TIM and these single variables has suggested a—
365 ---- weaker relationship (P=0.02) was found between---
400 Distinct growth curves (increased slopes and final abundances)--
445 --- composition of the eukaryotic community---
451 --- eukaryotes versus autotrophs and heterotrophs in the ocean.---
458 ---- therefore increase TIM. Similarly, deeper---
497 ---- the important role light plays for determining—
510 --- (such as ChrysoH) perform better under both low light and low nutrient—
531 --- Combining various approaches including in situ grazing experiments,--
637-538 --- oceans, phagotrophic mixotrophy is an efficient trophic strategy that small eukaryotes have evolved to survive nutrients impoverishment, deepening--
Also, I believe that the authors need to more carefully check that all data cited in the text are consistent with the same data shown in the figures and tables.
I noticed on page 6, Line 250, the authors state
“--- and over stable heterotrophic bacterial abundances (4.3-5.3 ×105 cells mL-1), However when I look in Table 1, the range is 3.9-5.3 ×105 cells mL-1; unless I have misinterpreted their text. There appear to be a few other small differences or inconsistencies such as this at places (?).
Some citations to the published literature are not properly formatted. There should be no comma after authors names before a parenthetical date. For example, Pesant et al., (2015), Picheral et al., (2014), Ardyna et al., (2017); Ibarbalz et al., (2019); Faure et al., (2019); Karlusich et al., (2022) etc.
should be
Pesant et al. (2015), Picheral et al. (2014), Ardyna et al. (2017); Ibarbalz et al. (2019); Faure et al. (2019); Karlusich et al. (2022) etc.
Also, throughout the manuscript, the authors do not properly use subscripts for the stoichiometry of the compounds. For example (P. 3, Line 135), the authors have not used numerical subscripts for the compounds (and also elsewhere throughout the manuscript)
“ --- Chl a, oxygen, nutrients (NO3+NO2, NO2, PO4, SiO4), PH, total alkalinity,--- “
Also, PH should be pH.
Comments on the Quality of English LanguageThe manuscript is nicely organized, but there are some corrections in use of English as I have recommended to the authors.
Author Response
To the Editor:
−We thank the two reviewers for their insightful comments. The majority of the suggestions, if not all, have been taken into account and incorporated into the revised manuscript (detailed answers were listed in the section below). We have made some additional minor revisions to improve the writing quality and the soundness of conclusions.
Reviewer 2
Overall, this is an interesting exploration of the prevalence and preferred niche of small eukaryotes with a focus on the role of mixotrophs in the global ocean. The authors merge data from the Tara Oceans eukaryotic 18S rRNA gene dataset and environmental variables to conduct large- scale field analysis, and isolates-based culture experiments to verify growth and nutritional resource relationships for representative taxa. Although the authors made some simplifying assumptions and adjustments in the data to permit a combined analysis of the total data set (which they acknowledge in the text and discussion), the overall results are informative and in many cases comport with prior published results on some of the key relationships that the authors report in their findings. In general, the research illustrates the value of this kind of merging of diverse data sources to provide a richer interpretation of open ocean dynamic processes.
−We are glad Reviewer 2 enjoyed reading this manuscript, and we are thankful for his/her suggestions to improve the quality of manuscript. We have particularly examined potential issues regarding the final dataset used in the manuscript. Indeed, something strange happened while merging our original dataset into the final supplementary tables (dataset S3) which has cause duplication of some of the rows/lines. All datasets have now been corrected in the revised manuscript. We have also approved, and incorporated all th rest suggestions into the new manuscript (marked with yellow highlights) raised by the Reviewer 2 as listed below.
Corrections for the authors to consider:
Line Correction
12 Unicellular eukaryotes that are capable of phago-mixotrophy in the ocean compete for inorganic nutrients and light----
Answers:We have made changes in the new abstract (highlighted colors).
18-20 i---- field analysis, and isolates-based culture experiments to verify growth and nutritional resource relationships for representative taxa. The field observations showed that compared to autotrophic and heterotrophic eukaryotes,----
Answers:We agree with the reviewer’s comments and have revised the first sentence, but the second sentence has been removed in the revised manuscript.
27 This study revealed a close relationship between---
Answers:We have made changes accordingly (highlighted colors).
39-40 --- in fact mixotrophs, phagotrophic mixotrophs particularly, that are capable of both—
Answers:We have made changes accordingly (highlighted colors).
50 mixotrophs and autotrophs compete for (inorganic) nutrients---
Answers:We have made changes accordingly (highlighted colors).
60 -- this universal mixotrophic trait; that is, photosynthesis and/or--
Answers:We have removed this sentence from the revised manuscript.
72 With climate change, evidence has shown that---
Answers:We have made changes accordingly (highlighted colors).
86 --- each lineage/species and assess relationship between---
Answers:We have made changes accordingly (highlighted colors).
104-105 Bathymetry depths varied from 13 to 5964 km and distance to coast was between 0.9-1404 km.
Answers:We have made changes accordingly (highlighted colors).
107 We built an environmental context dataset for further statistical analysis---
Answers:We have made changes accordingly (highlighted colors).
139 --- distance to coasts, sampling depths and bathymetry depths were---
Answers:We have made changes accordingly (highlighted colors).
157 Eukaryotic communities are known to have---
Answers:We have made changes accordingly (highlighted colors).
167-168 ---chloro-phytes, etc.). Finally, a correction factor (C.F.) for each group was calculated---
Answers:We have made changes accordingly (highlighted colors).
170 --- other small flagellates, respectively.
Answers:We have made changes accordingly (highlighted colors).
172 --- cell abundances by dividing gene numbers with C.F.—
Answers:We have made changes accordingly (highlighted colors).
174 --- were also assessed occasionally for comparison purposes.
Answers:We have made changes accordingly (highlighted colors).
179 --- each lineage/species (when possible) into three trophic groups,---
Answers:We have made changes accordingly (highlighted colors).
182-183 ---- (pigmented protists with no available evidence of phagocytosis) by searching evidence among published databases and references---
Answers:We have made changes accordingly (highlighted colors).
184-185 ---- at lineage or species level, their---
Answers:We have made changes accordingly (highlighted colors).
189 Can the authors please check this reported dataset umber ---- a final trophic composition dataset with 58,001,584 sequences----
Answers:We have checked those numbers and made corrections accordingly (moved to Page 5, line 188-190). Additional conclusions about sequences and trophic compositions were added into Page 6, line 217-219.
191-192 --- methodology, and address it further—
Answers:We have made changes accordingly (highlighted colors).
200 Surface datasets (SUR) were used---
Answers: The abbreviations for SUR and SUR&CML were now added into line 128-129.
209-210 -- that explained the environmental gradients from both---
Answers:We have made changes accordingly (highlighted colors).
212 -- extracted, TIg indices were plotted—
Answers:We have made changes accordingly (highlighted colors).
253-254 The higher nutrient concentrations in low latitude regions were possibly due to --
Answers:We have made changes accordingly (highlighted colors).
256-257 Sea was sampled in winter. Three trophic indices of TIM,==
AUTHORS please note you have misspelled principal component analysis throughout the manuscript. Your misspelled wording is printed as “principle component analysis –”
Answers:We thank the reviewer for pointing out this spelling error and have made changes throughout the manuscript (highlighted colors).
360 --- between TIM and these single variables has suggested a—
Answers:We have made changes accordingly (highlighted colors).
365 ---- weaker relationship (P=0.02) was found between---
Answers:We have made changes accordingly (highlighted colors).
400 Distinct growth curves (increased slopes and final abundances)--
Answers:We have made changes for the revised manuscript (highlighted colors).
445 --- composition of the eukaryotic community---
Answers:We have made changes accordingly (highlighted colors).
451 --- eukaryotes versus autotrophs and heterotrophs in the ocean.---
Answers:We have made changes accordingly (highlighted colors).
458 ---- therefore increase TIM. Similarly, deeper---
Answers:We have made changes accordingly (highlighted colors).
497 ---- the important role light plays for determining—
Answers:We have made changes accordingly (highlighted colors).
510 --- (such as ChrysoH) perform better under both low light and low nutrient—
Answers:We have made changes accordingly (highlighted colors).
531 --- Combining various approaches including in situ grazing experiments,--
Answers:We have made changes accordingly (highlighted colors).
637-538 --- oceans, phagotrophic mixotrophy is an efficient trophic strategy that small eukaryotes have evolved to survive nutrients impoverishment, deepening--
Answers:We have made changes accordingly (highlighted colors).
Also, I believe that the authors need to more carefully check that all data cited in the text are consistent with the same data shown in the figures and tables.
I noticed on page 6, Line 250, the authors state
“--- and over stable heterotrophic bacterial abundances (4.3-5.3 ×105 cells mL-1), However when I look in Table 1, the range is 3.9-5.3 ×105 cells mL-1; unless I have misinterpreted their text. There appear to be a few other small differences or inconsistencies such as this at places (?).
Answers:All data were examined carefully and corrections were made accordingly.
Some citations to the published literature are not properly formatted. There should be no comma after authors names before a parenthetical date. For example, Pesant et al., (2015), Picheral et al., (2014), Ardyna et al., (2017); Ibarbalz et al., (2019); Faure et al., (2019); Karlusich et al., (2022) etc.
should be
Pesant et al. (2015), Picheral et al. (2014), Ardyna et al. (2017); Ibarbalz et al. (2019); Faure et al. (2019); Karlusich et al. (2022) etc.
Also, throughout the manuscript, the authors do not properly use subscripts for the stoichiometry of the compounds. For example (P. 3, Line 135), the authors have not used numerical subscripts for the compounds (and also elsewhere throughout the manuscript)
“ --- Chl a, oxygen, nutrients (NO3+NO2, NO2, PO4, SiO4), PH, total alkalinity,--- “
Also, PH should be pH.
Answers:We thank the reviewer for the detailed comments, and have carefully made all changes accordingly throughout the manuscript (highlighted colors).
Comments on the Quality of English Language
The manuscript is nicely organized, but there are some corrections in use of English as I have recommended to the authors.
